# Siglec-6 Signaling Uses Src Kinase Tyrosine Phosphorylation and SHP-2 Recruitment

**DOI:** 10.3390/cells11213427

**Published:** 2022-10-29

**Authors:** Adrianne L. Stefanski, Michael D. Renecle, Anita Kramer, Shilpi Sehgal, Purnima Narasimhan, Kristen K. Rumer, Virginia D Winn

**Affiliations:** 1Department of Obstetrics and Gynecology, School of Medicine, University of Colorado, Aurora, CO 80309, USA; 2Department of Obstetrics and Gynecology, School of Medicine, Stanford University, Stanford, CA 94305, USA

**Keywords:** Siglec-6, trophoblast, preeclampsia, phosphorylation, signal transduction

## Abstract

Preeclampsia is a pregnancy-specific disorder involving placental abnormalities. Elevated placental Sialic acid immunoglobulin-like lectin (Siglec)-6 expression has been correlated with preeclampsia. Siglec-6 is a transmembrane receptor, expressed predominantly by the trophoblast cells in the human placenta. It interacts with sialyl glycans such as sialyl-TN glycans as well as binds leptin. Siglec-6 overexpression has been shown to influence proliferation, apoptosis, and invasion in the trophoblast (BeWo) cell model. However, there is no direct evidence that Siglec-6 plays a role in preeclampsia pathogenesis and its signaling potential is still largely unexplored. Siglec-6 contains an immunoreceptor tyrosine-based inhibitory motif (ITIM) and an ITIM-like motif in its cytoplasmic tail suggesting a signaling function. Site-directed mutagenesis and transfection were employed to create a series of Siglec-6 expressing HTR-8/SVneo trophoblastic cell lines with mutations in specific functional residues to explore the signaling potential of Siglec-6. Co-immunoprecipitation and inhibitory assays were utilized to investigate the association of Src-kinases and SH-2 domain-containing phosphatases with Siglec-6. In this study, we show that Siglec-6 is phosphorylated at ITIM and ITIM-like domains by Src family kinases. Phosphorylation of both ITIM and ITIM-like motifs is essential for the recruitment of phosphatases like Src homology region 2 containing protein tyrosine phosphatase 2 (SHP-2), which has downstream signaling capabilities. These findings suggest Siglec-6 as a signaling molecule in human trophoblasts. Further investigation is warranted to determine which signaling pathways are activated downstream to SHP-2 recruitment and how overexpression of Siglec-6 in preeclamptic placentas impacts pathogenesis.

## 1. Introduction

Preeclampsia is a pregnancy-specific disorder affecting 3–8% of human pregnancies [1]. The placenta plays a central role in the development of preeclampsia. Fetal trophoblast cells are key players that establish the placenta by invading the hormonally altered maternal endometrium or decidua (interstitial invasion), as well as by remodeling maternal vasculature (endovascular invasion) in the decidua basalis region of the placenta. Trophoblast cells also form and line the chorionic villi which compose the majority of the placenta. Chorionic villi are the site for transport between the mother and the fetus and are responsible for the extensive production of hormones by the placenta. Defects of the trophoblasts within both the decidua basalis and chorionic villous regions of the placenta have been associated with preeclampsia pathogenesis [2].

Our investigations into differentially expressed genes at the maternal-fetal interface revealed increased expression of Siglec-6 (Sialic acid immunoglobulin-like lectin-6) in the trophoblasts of the placentas from severe, preterm preeclampsia compared to normotensive preterm controls [3]. Siglec-6 is a transmembrane sialic acid-binding protein and a member of the immunoglobulin superfamily [4]. Siglec-6 is expressed in both the syncytiotrophoblast and cytotrophoblast cells of the chorionic villi and the extravillous trophoblasts of the decidua basalis in normal human placenta. It is also expressed in B-cells of all studied primates; however, its placental expression is unique to humans [5]. Intriguingly, preeclampsia is a complication primarily observed in human pregnancies [6]. Although Siglec-6 overexpression has been well correlated with preeclampsia [7], the Siglec-6 signaling mechanism in placental trophoblast cells has not been examined.

Despite extensive characterization of cell and tissue expression of Siglec-6 [5,8,9], a CD33-related Siglec [4], little is known about the signaling capability of this transmembrane protein. Members of the CD33-related Siglecs have conserved domains and amino acid residues including (1) a terminal v-set immunoglobulin domain containing an arginine residue critical for binding to sialic acids, (2) additional C2-set extracellular immunoglobulin domains, (3) a transmembrane domain, and (4) intracellular domain with ITIM and ITIM-like motifs [4]. Many members of this protein family have known intracellular signaling functions involving tyrosine phosphorylation at one or both inhibitory tyrosine residues. For many structurally similar Siglecs, Src-family tyrosine kinases have been shown to be responsible for Siglec ITIM phosphorylation. Phosphorylated tyrosines within Siglec ITIMs are known to recruit src homology (SH)-2 domain-containing phosphatases including SHP-1 and SHP-2 [10,11]. However, it is currently unknown if Siglec-6 becomes tyrosine phosphorylated and has intracellular signaling capability. We hypothesized that trophoblast-expressed Siglec-6 becomes phosphorylated at ITIM and ITIM-like tyrosines leading to the recruitment of phosphatases known to have downstream signaling capabilities.

To determine if Siglec-6 possesses intracellular signaling capabilities, we used an immortalized first-trimester human trophoblast line (HTR-8/SVneo) stably expressing Siglec-6. Site-directed mutagenesis of specific residues and transfection were employed to create a series of Siglec-6 expressing HTR-8/SVneo cell lines to determine the contribution of each of these residues to Siglec-6 intracellular signaling capabilities. Pervanadate, known to prevent tyrosine dephosphorylation in other CD33 family Siglecs, was utilized to inhibit tyrosine de-phosphorylation. Co-immunoprecipitation (Co-IP) and immunoblotting were then used to detect tyrosine-phosphorylated Siglec-6. Further, we utilized first-trimester placental tissue to extend our investigations to include endogenously expressed Siglec-6. Finally, we investigated the association of Src-kinases and SH-2 domain-containing phosphatases in Siglec-6 using kinase inhibitors and co-immunoprecipitation. This work demonstrates the signaling potential of the uniquely expressed placental Siglec-6 suggesting it plays a functional role in normal trophoblast biology and its overexpression may contribute to preeclampsia pathogenesis.

## 2. Materials and Methods

### 2.1. Placental Tissue Collection

Placental collection was approved by the University of Colorado Multiple Institution Review Board (COMIRB #06-1098). All patients consented to donate placental tissue. First-trimester placentas from elective pregnancy terminations between 6–10 weeks of gestation were collected. Exclusion criteria for this study included evidence of infection, hydropic changes, and known genetic or fetal anomalies. Gestational age was based on last menstrual period (LMP) and confirmed by ultrasound. Tissue was washed extensively in cold phosphate-buffered saline (PBS) and used for immunoprecipitation (IP) (as described below).

### 2.2. Cell Lines

The immortalized first-trimester human trophoblast cell line, HTR-8/SVneo (established by immortalizing trophoblast cells via transfection with SV-40 containing plasmid [12]), was cultured in RPMI 1640 supplemented with 5% fetal bovine serum, 100 U/mL penicillin, and 100 µg/mL streptomycin. HTR-8/SVneo cells were maintained in monolayer cultures on plastic at 37 °C in 5% CO_2_, 95% air, and 95% humidity. Jurkat cells, which are an immortalized human T lymphocyte cell line [13], were grown in RPMI 1640 with 10% cosmic calf serum, 10 mM Hepes, 100 units/mL penicillin, and 100 μg/mL streptomycin. Jurkat cells were grown at 37 °C in 5% CO_2_, 95% air and 95% humidity in suspension on plastic T-75 flasks for use as a positive control for SHP-1 and SHP-2 protein.

### 2.3. Expression Plasmids and Site-Directed Mutagenesis

From the pCDNA3.1+ (Invitrogen) vector full-length Siglec-6 cDNA was sub-cloned into the pCDH-CMV-MCS-EF1-GFP-T2A-Puro (pCDH, System Biosciences, Palo Alto, CA, USA) vector using EcoRI and NotI restriction sites as previously described [7,8]. Empty pCDH plasmid was used as a negative control. Individual mutants of the full-length Siglec-6 constructs were generated by site-directed mutagenesis using the QuickChange II Mutagenesis Kit (Agilent Technologies, Santa Clara, CA, USA) according to the manufacturer’s protocols. Arginine to Alanine (R111A) mutation that disables sialic acid ligand binding was introduced in the variable set domain of the Siglec-6 receptor and tyrosine to phenylalanine mutation was created in membrane-proximal ITIM (Y413F) and distal ITIM-like (Y433F) domain. A double mutant (YFYF) with both Y413F and Y433F mutations was also generated. Serine 1521 was mutated to a STOP codon to generate an intracellularly truncated version of Siglec-6 (ICTrunc) to use as a negative control. A cartoon representing the structural domains of Siglec-6 protein with specific site-directed mutations created in these domains is shown in Figure 1. Site-directed mutagenesis was performed using a set of mutagenic primers (IDT, Coralville, IA, USA) listed in Table 1.

The resultant DNA constructs were transformed in XL-10 Gold (Agilent Technologies, Santa Clara, CA, USA) or DH5α (Invitrogen, Waltham, MA, USA) competent cells which were then plated onto LB Agar plates containing 100 mg/mL ampicillin overnight. Individual colonies were selected and used to inoculate LB broth incubated at 37 °C with shaking. Plasmid DNA was isolated using QIAprep Spin Miniprep kit (Qiagen, Hilden, Germany). Successful introduction of the desired mutations was confirmed by DNA sequencing using a set of sequencing primers shown in Table 2.

### 2.4. Generation of Stably Expressing Full-Length and Mutated Siglec-6 HTR-8/SVneo Cell Lines

HTR-8/SVneo cells were transfected with pCDH plasmid DNA containing no insert, full-length Siglec-6 or one of the mutated versions of Siglec-6 (R111A, Y413F, Y433F, YFYF and ICTrunc,) using Lipofectamine 2000 (Invitrogen, Waltham, MA, USA) as per the manufacturer’s protocol. 24 h post-transfection, transfection efficiency was determined by measuring cellular green fluorescent protein (GFP)-expression using a Nikon eclipse 80i fluorescent microscope. 48 h post-transfection cells were selected using 1 µg/mL puromycin, which was experimentally determined to be the minimum dose necessary to kill 100% of non-transfected HTR-8/SVneo cells.

### 2.5. Analysis of Tyrosine Phosphorylation of Siglec-6 and Interaction with SHP-1 and SHP-2 in HTR-8/SVneo Cells and Chorionic Villi from First-Trimester Placenta

Co-Immunoprecipitation-HTR-8/SVneo cells grown on plastic plates (Thermo Scientific, Waltham, MA, USA), were treated with 100 µM pervanadate (100 µM sodium orthovanadate, 3.4% hydrogen peroxide) or PBS for 10 min at 37 °C, washed with PBS, and lysed with lysis buffer (PBS with 1% Triton-X-100, 1% Protease Inhibitor Cocktail (Sigma, St. Louis, MO, USA, P8340, 1 mM EDTA, 1 µM NaF, 2 mM NaVO_4_). Similarly, chorionic villi dissected from first-trimester placental tissue were either treated with pervanadate or PBS for 10 min at 37 °C. The chorionic villi tissue was pulverized using a 15 mL closed tissue grinder system and then the cells were lysed using a cell lysis buffer. Anti-Siglec-6 antibody (R&D Systems, Minneapolis, MN, USA) was incubated with Sepharose A beads ((GE Healthcare) (1 µg antibody per 10 µL beads) for 1 h at 4 °C with rotation. The lysates prepared from HTR-8/SVneo cells and chorionic villous tissue were separately centrifuged at 5900× *g* for 10 min at 4 °C, precleared with Sepharose A beads and then incubated overnight at 4 °C with anti-Siglec-6 antibody bound to Sepharose A beads. The beads were washed after incubation, three times in lysis buffer and then heated in lamelli buffer (Bio-Rad, Shinagawa City, Tokyo) with β-mercaptoethanol (Sigma, St. Louis, MO, USA) for 10 min at 95 °C and then chilled on ice.

Immunoblotting—Protein or immunoprecipitated lysates were immunoblotted using standard reducing SDS-PAGE techniques. Briefly, equal volumes of protein or (co-) immunoprecipitated lysates were loaded on precast 10% Tris-HCl criterion gels (Bio-Rad, Shinagawa City, Tokyo). Because Siglec-6, SHP-1and SHP-2 migrate to a similar position on an SDS-PAGE gel, one gel was run per antibody for each replicate (*n* = 2–4). The resolved protein bands were transferred to either polyvinylidene difluoride (PVDF) (BioRad, Shinagawa City, Tokyo) or nitrocellulose (Bio-Rad, Shinagawa City, Tokyo) membrane. All membranes were incubated overnight with primary antibody at 4 °C on a rocking platform. Anti-Siglec-6 1:500 (R & D Systems, Minneapolis, MN, USA), anti-phosphotyrosine (1:1000) (Millipore, 4G10) anti-SHP-1, 1:200 (Santa Cruz), and anti-SHP-2 1:200 (Santa Cruz) primary antibodies were used. Anti-Siglec-6 blots were blocked with 5% blocking grade milk in TBST (BioRad, Shinagawa City, Tokyo); anti-SHP-1, anti-SHP-2 and anti-phosphotyrosine blots were blocked with 3% BSA in TBST. The bound antibody was detected with the appropriate horseradish-peroxidase (HRP)-conjugated secondary antibodies (Siglec-6, anti-goat; SHP-1/SHP-2, anti-rabbit; anti-phosphotyrosine, anti-mouse from Rockland) and visualized using western-lightning chemiluminescent substrate (PerkinElmer, Waltham, MA, USA) as per manufacturer’s instructions.

### 2.6. Src Kinase Inhibition Studies

HTR-8/SVneo cells expressing empty pCDH plasmid or Siglec-6 were grown in monolayers on plastic plates and treated with 5 µM pervanadate (experimentally determined dose sufficient to allow detection of phosphorylated tyrosine by immunoprecipitation), followed by immunoblotting with anti-phosphotyrosine antibody. Cells were treated with 5 µM pervanadate for 10 min alone or in combination with the Src-kinase inhibitor Pyrazolopyrimidine (PP2) (100 µM, Tocris, Bristol, UK). This experiment was also performed using PP2 as a 1 h pretreatment, followed by the addition of 5 µM pervanadate for 10 min. HTR-8/SVneo cells expressing pCDH and Siglec-6 were pretreated for 1 h with or without 10 µM Dasatinib (Cell Signaling Technology, Danvers, MA, USA), followed by addition of 5 µM pervanadate for 10 min. Following treatment, cells were washed, lysed, immunoprecipitated, and immunoblotted as described above.

## 3. Results

### 3.1. pCDH Vector-Mediated Expression of Siglec-6 in HTR-8/SVneo Cells Allows for In Vitro Detection of Siglec-6 Tyrosine Phosphorylation

To investigate the tyrosine phosphorylation-mediated signaling capabilities of Siglec-6 expressed in trophoblasts we utilized an in vitro model system. We transfected the immortalized first-trimester human trophoblast line, HTR-8/SVneo, that does not express Siglec-6 under normal culture conditions with a pCDH vector containing no insert (pCDH) or a Siglec-6 cDNA (Siglec-6). Successful transfection of pCDH or Siglec-6 was confirmed by RT-qPCR (Figure 2a) and by visualizing Green Fluorescent Protein (GFP) expression in transfected HTR8/SVneo cells (Figure 2b). To investigate if Siglec-6 can become tyrosine phosphorylated, pCDH and Siglec-6 expressing HTR-8/SVneo cells were treated with and without the phosphatase inhibitor pervanadate, immunoprecipitated, and the resulting lysates were immunoblotted in duplicate using anti-Siglec-6 and anti-phosphotyrosine (4G10) antibodies (Figure 2c). As expected, due to the lack of Siglec-6 expression in HTR-8/SVneo cells under normal conditions; no Siglec-6 was immunoprecipitated from pCDH transfected HTR-8/SVneo cells (Figure 2c, upper panel, Lane 1 and 2). A specific band of 60kDa corresponding to Siglec-6 was observed in the immunoprecipitated lysate from Siglec-6 transfected HTR-8/SVneo cells. When treated with pervanadate, immunoprecipitated Siglec-6 shows an upward gel shift consistent with phosphorylation on SDS-PAGE gel under reducing conditions. (Figure 2c, upper panel, Lane 3 and 4). Anti-phospho-tyrosine antibody confirmed the mobility shift was due to tyrosine phosphorylation of Siglec-6, (Figure 2c, Lower panel, Lane 3 and 4). Siglec-6 is phosphorylated on tyrosine residues and rapidly dephosphorylates in normal culture conditions without pervanadate.

### 3.2. Human Placental Siglec-6 Is Tyrosine-Phosphorylated

To confirm our in vitro finding that Siglec-6 tyrosine phosphorylation recapitulates endogenously expressed Siglec-6 in the human chorionic villi, we used first-trimester human placental tissue given its known high expression of Siglec-6. [8]. Siglec-6 was found to be tyrosine-phosphorylated in pervanadate-treated chorionic villi as shown by the anti-phosphotyrosine immunoreactivity of immunoprecipitated Siglec-6 (Figure 3).

### 3.3. Site-Directed Mutations of Siglec-6 Reveal Intracellular ITIM and ITIM-like Tyrosine Phosphorylation

To determine which conserved structural domains of Siglec-6 are critical for the Siglec-6 tyrosine phosphorylation, we created mutant forms of Siglec-6 using site-directed mutagenesis. (Figure 4). Both ITIM tyrosine Y413 and the ITIM-like Y433 were mutated to phenylalanine individually (Y413F and Y433F) and in combination (YFYF). As an additional negative control, we generated a truncated version of Siglec-6 that retained the transmembrane and extracellular domains, but no longer included the cytoplasmic tail, by changing serine1521 to a stop codon. Siglec-6 contains a conserved arginine residue (R111), which is critical for Siglec-Sialic acid interactions. Thus, arginine 111 was mutated to alanine (R111A) to determine if a mutation in the ligand binding domain impacted the tyrosine phosphorylation. HTR-8/SVneo cells stably transfected with pCDH, Siglec-6, Y413F, Y433F, YFYF, ICTrunc and R111A were treated with PBS or pervanadate and immunoprecipitated with anti-Siglec-6 antibody. Lysates were analyzed with immunoblots probed with anti-Siglec-6 and anti-phosphotyrosine antibodies (Figure 3b). The immunoblot demonstrates a Siglec-6 band corresponding to 60 kDa molecular weight except for ICTrunc construct which lacks the intracellular domain and shows a downward shift on the gel. The anti-phosphotyrosine immunoblot confirms phosphorylation of the ITIM (Y413) and ITIM-like (Y433) tyrosines. Although not quantitative, the mutation of either tyrosine (Y413F or Y433F) dramatically reduced the amount of Siglec-6 tyrosine phosphorylation signal as compared to Siglec-6 with neither tyrosine mutation. As expected, both the YFYF and ICTrunc mutations resulted in the complete abrogation of Siglec-6 tyrosine phosphorylation. Additionally, HTR-8/SVneo cells expressing R111A were not prevented from pervanadate-induced Siglec-6 tyrosine phosphorylation.

### 3.4. Dually Tyrosine Phosphorylated Siglec-6 Associates with SHP-2

To determine if SHP-1 and/or SHP-2, the SH-2 domain-containing phosphatases recruited to phosphorylated tyrosines, are expressed in HTR-8/SVneo cells, we treated HTR-8/SVneo with or without pervanadate and the lysates were immunoblotted and probed with anti-SHP-1 and anti-SHP-2 antibodies. Jurkat cells were used as a positive control. Only SHP-2 was found to be expressed in HTR-8/SVneo cells at detectable levels regardless of treatment condition (Figure 5a). Next, to determine if SHP-2 is recruited and which tyrosine phosphorylated sites are involved, Siglec-6 was co-immunoprecipitated from HTR-8/SVneo cells expressing the various Siglec-6 mutants. Immunoblots were probed with anti-Siglec-6 and anti-SHP-2 antibodies (Figure 5b). Full-length wildtype Siglec-6 associated with SHP-2 in trophoblasts following pervanadate treatment. The recruitment of SHP-2 to Siglec-6 was abrogated when either or both intracellular tyrosines were mutated to phenylalanine suggesting that SHP-2 recruitment requires phosphorylation at both tyrosine sites. A mutation in the sialic acid-binding region of Siglec-6 (R111A) did not affect the ability of phosphorylated Siglec-6 to recruit SHP-2 following pervanadate treatment.

### 3.5. Siglec-6 Phosphorylation and Recruitment of SHP-2 Was Prevented by Treatment with Inhibitors of Src-Family Kinases

To determine which kinase phosphorylate Siglec-6, HTR-8/SVneo cells expressing Siglec-6 were treated with Src and Abl kinase inhibitors in combination with pervanadate. Since SHP-2 was found to only be strongly recruited to dually tyrosine-phosphorylated Siglec-6, only HTR-8/SVneo cells expressing non-mutated Siglec-6 were included in this experiment. We pre-treated non-mutated Siglec-6 expressing HTR-8/SVneo cells with the Src and Abl kinase inhibitor, Dasatinib (DAS) followed by treatment with 5 uM pervanadate. DAS treatment led to a reduction of Siglec-6 phosphorylation and inhibition of SHP-2 recruitment (Figure 6a). Next, we co-treated Siglec-6 expressing HTR-8/SVneo cells with a specific Src kinase inhibitor, pyrazolopyrimidine (PP2), in combination with pervanadate. SHP-2 recruitment was also abrogated by PP2 treatment (Figure 6b). Together, these results suggest that Siglec-6, expressed in trophoblasts, is phosphorylated by Src-family kinases and that the resultant phosphorylation of Siglec-6 at both ITIM and ITIM-like tyrosine residues leads to the recruitment of SHP-2.

## 4. Discussion

We demonstrate that human placental Siglec-6 expressed in chorionic villi is tyrosine phosphorylated in the presence of the phosphatase-inhibitor pervanadate. In vitro inhibitor studies reveal that Src-family kinases are responsible for phosphorylating Siglec-6 at its phospho-tyrosine sites. Further, the SH-2 domain-containing phosphatase, SHP-2, is recruited to tyrosine phosphorylated Siglec-6. By mutating both Siglec-6 intracellular tyrosine residues, we demonstrate that each can be phosphorylated independently of the other. However, phosphorylation of both the ITIM Y413 and ITIM-like Y433 are necessary for the robust recruitment of SHP-2 to Siglec-6. Siglec-6 phosphorylation is only detectable in the presence of a phosphatase inhibitor, suggesting that the activity of SH-2 phosphatases, including SHP-2, predominates over the phosphorylation of Siglec-6 by Src-family kinases. These data support our hypothesis that trophoblast-expressed Siglec-6 becomes phosphorylated at the ITIM and ITIM-like tyrosine leading to the recruitment of phosphatases with downstream signaling capabilities. Additionally, these data add to the existing body of evidence that Siglec-6 is a functional CD33-related receptor with intracellular signaling potential.

In the absence of tyrosine phosphorylation, recruitment of SHP-2 to Siglec-6 was dramatically decreased, as demonstrated in cells expressing mutated ITIM and ITIM-like domains. This is consistent with evidence from CD33 in which mutation of either the ITIM or ITIM-like tyrosine residue reduced recruitment of SHP-2 [14]. The downstream signaling consequences of Siglec-6 phosphorylation and recruitment of SHP-2 are yet to be demonstrated. SHP-2 participates in a variety of intracellular signaling functions, but its actions are not yet well described in the placenta. Interestingly, SHP-2 is known to participate in two of the major signaling pathways thought to regulate trophoblast function: the receptor-associated tyrosine kinase mitogen-activated protein kinase (RTK-MAPK) and the Janus kinase-signal transducers and activators of transcription (Jak-Stat) [15]. SHP-2 activation of MAPK members, extracellular signal-related kinases 1 and 2 (ERK), have been shown to promote cell growth and differentiation in many systems. For normal placental development, ERK is required for normal growth, differentiation, and invasion of trophoblasts [16,17]. Interestingly, the expression pattern of active ERK in early pregnancy is similar to that of Siglec-6 in villous cytotrophoblasts, with high trophoblastic expression at the beginning of gestation [8]. In trophoblast stem cells, the SHP-2/SFK/Ras/Erk signaling pathway is necessary for the FGF4-mediated stem cell survival and maintenance [18].SHP2 is also known to regulate IGF-mediated proliferation in human trophoblast cells [19,20].

SHP-2 is also known to interact with members of the Jak-Stat pathway and may be required for the activity of STAT3, STAT5, and Jak2 [15,21,22]. In trophoblasts, STAT3 activation is positively correlated with invasiveness in trophoblast cells [23]. Increased STAT3 expression has also been associated with villous trophoblast differentiation [24]. Further investigation is critical to gain insights into the signaling pathways that are activated downstream of SHP-2 following dephosphorylation of Siglec-6, including investigation of SH-2 domain-containing proteins such as SOCS3 and SHIP. This investigation would bring to light, a novel mechanism of STAT3 activation mediated by Siglec-6 apart from the known LIF (Leukemia Inhibitory Factor)-mediated STAT3 activation in trophoblast cells. In addition to regulating downstream molecules, SHP-2 may also activate Src-family kinases indirectly [25].

One possible cellular consequence of aberrantly increased Siglec-6 in preeclampsia is the recruitment of Src-family kinases and SHP-2 to Siglec-6, thereby altering the signaling of other membrane-associated receptors containing tyrosine signaling motifs such as growth factors and other receptor tyrosine kinases. Specifically, crosstalk between the RTK-MAPK and Jak-Stat pathways by multiple signaling molecules has been reported, including via SHP-2 [15]. By sequestering signaling proteins to areas of the membrane rich in Siglec-6, they would be spatially and temporally unavailable to interact with SH2 domain-containing proteins that utilize the same signaling molecule for their normal function, thus throwing off the balance of intracellular signals and ultimately altering cellular function.

Investigations of CD33 demonstrate that phosphorylation of the ITIM and ITIM-like tyrosine residues regulate the rate of CD33 internalization [26]. Although we did not directly address Siglec-6 internalization, the observation of phosphorylation of ITIM and ITIM-like tyrosines support the potential for Siglec-6 endocytosis following extracellular ligand binding as is currently thought to be a functional mechanism of all CD33-related Siglecs. Further investigation is needed to determine if Siglec-6 is internalized and whether internalization is tyrosine phosphorylation-dependent.

We show that Siglec-6 is phosphorylated at ITIM and ITIM-like domains by Src family kinases. Phosphorylation of both ITIM and ITIM-like motifs is essential for the recruitment of phosphatases like Src homology region 2 containing protein tyrosine phosphatase 2 (SHP-2), which has downstream signaling capabilities. This work demonstrates the signaling potential of the uniquely expressed placental Siglec-6 suggesting it plays a functional role in normal trophoblast biology. In addition, further investigation is warranted to understand how overexpression of Siglec-6 in placentas from preeclamptic pregnancies impacts trophoblast function.

## Figures and Tables

**Figure 1 cells-11-03427-f001:**
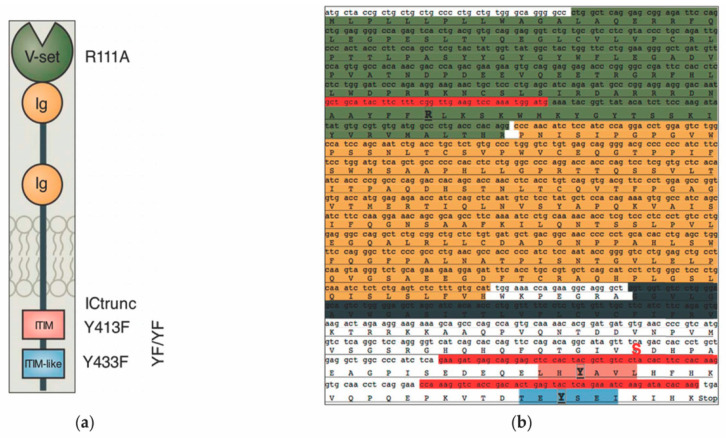
(**a**) A graphical representation of Siglec-6 representing the structural components of the proteins including the v-set sialic-acid binding domain (green), two extracellular immunoglobulins (Ig) domains (gold), the transmembrane domain (grey), and the intracellular ITIM (pink) and ITIM-like (blue) putative signaling motifs (adapted from [9]). (**b**) *SIGLEC6* gene sequence with primers used for mutagenesis (highlighted in red) and specific site-directed mutations that were created in the structural domains (highlighted in bold): Alanine to arginine mutation in the v-set domain (R111A), Tyrosine to phenylalanine mutations in the membrane-proximal ITIM (Y413F), distal ITIM-like (Y433F), and both (YFYF). A serine1521 to a STOP codon mutation was created to generate an intracellularly truncated version of Siglec-6 (ICTrunc) (highlighted in red and bold).

**Figure 2 cells-11-03427-f002:**
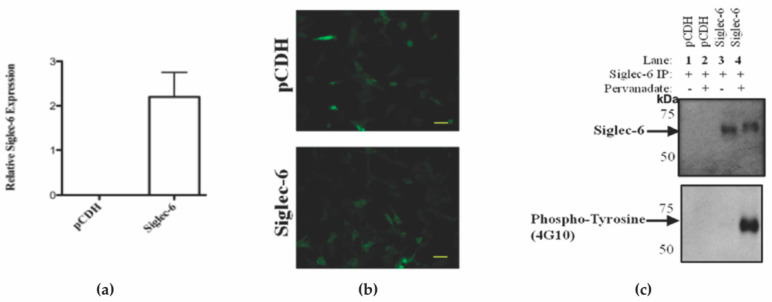
Siglec-6 Stably Expressed in HTR-8/SVneo Cells Undergoes Tyrosine Phosphorylation following pervanadate treatment. Successful transfection of pCDH or Siglec-6 in HTR-8/SVneo cells was confirmed by (**a**) qPCR showing Siglec-6 expression in HTR-8/SVneo cells transfected with pCDH or Siglec-6. (Data are depicted as Mean ± SEM; triplicates) and (**b**) visualizing cellular GFP expression. (yellow bar = 75 µm). (**c**) Upper panel: immunoblot for Siglec-6 of HTR-8/SVneo cells transfected with either pCDH vector alone or pCDH vector containing Siglec-6 cDNA insert. Lower panel: immunoblot for tyrosine phosphorylation of HTR-8/SVneo cells transfected with pCDH vector alone or containing Siglec-6 cDNA insert plus and minus pervanadate-treatment (*n* = 3).

**Figure 3 cells-11-03427-f003:**
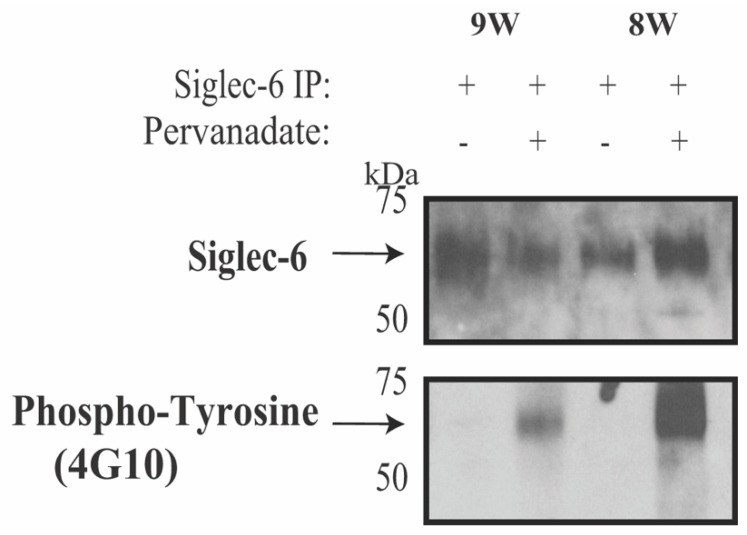
Siglec-6 undergoes Tyrosine Phosphorylation in First Trimester Human Placenta. First trimester human placental chorionic villus tissue (gestational age 8 and 9 weeks, 8W and 9W) was incubated with PBS alone or pervanadate. Tissue was then washed, pulverized, lysed, and immunoprecipitated with an anti-Siglec-6 antibody. The immunoprecipitates were analyzed by duplicate immunoblots probed with anti-Siglec-6 or anti-phosphotyrosine antibodies. A total of four human placental samples were used over two experiments with similar results.

**Figure 4 cells-11-03427-f004:**
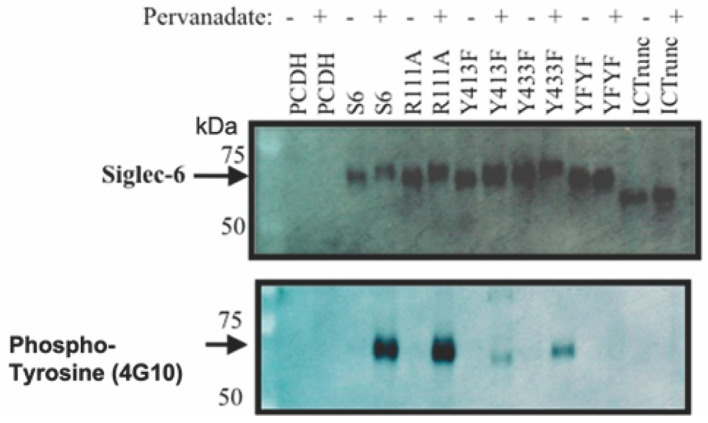
Siglec-6 ITIM and ITIM-like Tyrosine are Phosphorylated in HTR-8/SVneo Cells. pCDH, Siglec-6, and mutated forms of Siglec-6 were transfected into HTR-8/SVneo cells to generate stable cell lines. Cells were treated with PBS (control) or pervanadate, washed, lysed, and immunoprecipitated with anti-Siglec-6 antibody. Immunoprecipitates were run on duplicate immunoblots and probed with anti-Siglec-6 or anti-phosphotyrosine antibodies (*n* = 3).

**Figure 5 cells-11-03427-f005:**
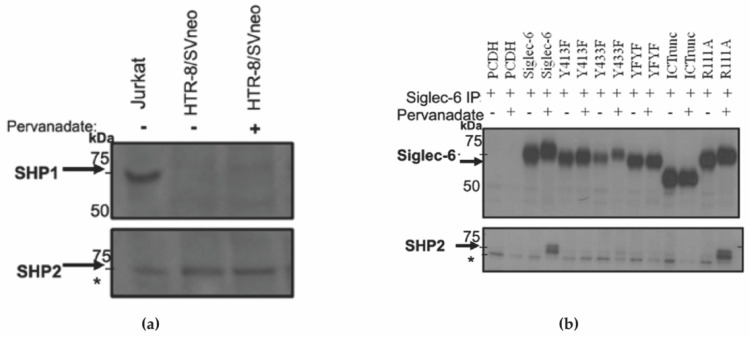
SHP-2 is Recruited to Siglec-6 Following Tyrosine Phosphorylation. (**a**) Whole cell lysates from Jurkat and HTR-8/SVneo treated with and without pervanadate were immunoblotted using anti-SHP-1 (upper panel) and anti-SHP-2 (lower panel) antibodies (**b**) pCDH, Siglec-6, and mutated Siglec-6 were transfected into HTR-8/SVneo cells to generate stable cell lines. Cells were treated with PBS (control) or 100 µM pervanadate, and lysates immunoprecipitated with anti-Siglec-6 antibody. Co-immunoprecipitates were run on duplicate immunoblots and probed with anti-Siglec-6 or anti-SHP-2 antibodies (*n* = 2–4). * A non-specific band was observed across all lanes.

**Figure 6 cells-11-03427-f006:**
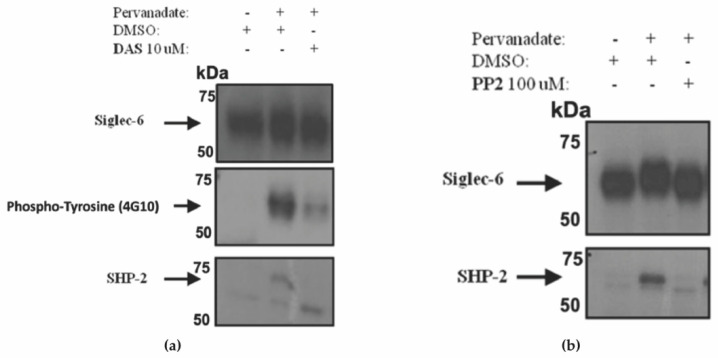
Inhibition of Src-kinases Prevents Siglec-6 Phosphorylation and SHP-2 Recruitment. HTR-8/SVneo cells stably transfected with Siglec-6 or pCDH control were treated with PBS (control) or pervanadate in conjunction with (**a**) one-hour pretreatment with DMSO or DAS. Anti-Siglec-6 immunoprecipitates were immunoblotted with anti-Siglec-6, anti-phosphotyrosine (4G10), and anti-SHP-2 antibodies. (**b**) HTR-8/SVneo lysates from cells pretreated with DMSO or PP2 were immunoprecipitated and immunoblotted with anti-Siglec-6 and anti-SHP-2 antibodies (*n* = 2–4).

**Table 1 cells-11-03427-t001:** Primers used for mutagenesis in this study.

Name	Sequence
R111A Forward	5′-GACAATGCTGCATACTTCTTT**GCC**TTGAAGTCCAAATGGATG-3′
R111A Reverse	5′-CATCCATTTGGACTTCAA**GGC**AAGAA GTA TGC AGC ATT GTC-3′
Y413F Forward	5′-GAAGATGAGCAGGAGCTCCAC**TTC**GCTGTCCTACACTTCCACAAG-3′
Y413F Reverse	5′-CTTGTGGAAGTGTAGGACAGC**GAA**GTG GAGCTCCTGCTCATCTTC-3′
Y433F Forward	5′-CCAAAG GTCACCGACACTGAG**TTC**TCAGAAATCAAGATACACAAG-3′
Y433F Reverse	5′-CTTGTGTATCTTGATTTCTGA**GAA**CTCAGTGTCGGTGACCTT TGG-3′
ICtrunc Forward	5′-CAGTTCCAGACAGGCATAGTT**TGA**GACCACCCTGCTGAGGCTGGC-3′
ICtrunc Reverse	5′-GCCAGCCTCAGCAGGGTGGTC**TCA**AACTATGCCTGTCTGGAACTG-3′

Note-Nucleotide changes introduced for generating each mutant are highlighted in bold.

**Table 2 cells-11-03427-t002:** Primers used for sequencing the DNA constructs.

Name	Sequence
CMV Forward	5′-CACGCTGTTTTGACCTCCATAGA-3′
Siglec-6 Forward	5′-GGAGAGAACATGGTACCTCTCAGT-3′
Siglec-6 Mid Forward	5′-GCTGCCCCCCACCTCCTGGGCCCC-3′
Siglec-6 Reverse	5′-AAGACACAAGGAGGAGACAGCCAT-3′

## Data Availability

All data generated or analyzed during this study are included in this published article.

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
