# Peer review of "Siglec-6 Signaling Uses Src Kinase Tyrosine Phosphorylation and SHP-2 Recruitment"

_cells, 2022, doi:10.3390/cells11213427_

Round 1
Reviewer 1 Report
1) Line 83. “First trimester placentas from elective pregnancy terminations between 5-8 weeks of gestation were collected. Exclusion criteria for this study included evidence of infection, hydropic changes, and known genetic or fetal anomalies. Gestational age was determined using standard ultrasound.” 5-8 weeks GA cannot be determined by ultrasound normally. It would be 8-13 weeks. I guess it might be defined from last menstrual period.
2) GENERAL: Line 374, “One possible cellular consequence of aberrantly increased Siglec-6 in preeclampsia is the recruitment of Src-family kinases and SHP-2 to Siglec-6, thereby altering the signaling of other membrane-associated receptors containing tyrosine signaling motifs such as growth factors and other receptor tyrosine kinases.”, Line 396, “In addition, further investigation is warranted to understand how overexpression of Siglec-6 in placentas from preeclamptic pregnancies impacts trophoblast function and preeclampsia pathogenesis.”, and so on: The authors must state clearly that which stage of preeclampsia do they wants to mimic by this study. 5-8 weeks GA is an extremely early stage of pregnancy without strong interaction with maternal immune cells. HTR8/SvNeo comes from around 10 weeks gestation, that interaction with uterine NK cells are suggested (though it is immortalized and some character may be changed).
In line 63, the authors wrote “We hypothesized that trophoblast-expressed Siglec-6 becomes phosphorylated at ITIM and ITIM-like tyrosines leading to the recruitment of phosphatases known to have downstream signaling capabilities.”. It might be revealed partially by this study, but what kind of pathogenesis by Siglec do they want to show? Less spiral artery remodeling by extravillous trophoblasts on 12-15weeks GA, or incomplete fusion of the syncytiotrophoblasts on <20w GA? Problems written before Lines 39 are not tried to be solved in this study or discussion.
Reviewer 2 Report
Major Revisions:
Figure 3, upper panel, Siglec-6 antibody signal is almost non-existing and the phosphotyrosine signal is diffused and does not represent a “single band”. A different blot image needs to be provided, and for the experiment to be repeated using villi from different placentas if not already done.
Minor Revisions:
1. All figures are missing the number of replicates (biological, not technical) for each experiment performed. Please add.
2. All IP figures are missing isotype control and input (I assume it was done), please show them in figures, at least in some, and at least with SIGLEC-6 transfected HTR8/SVneo.
3. The general quality of the immunoblots is not great. Some blots are better than others. I was wondering if the running conditions were consistent across experiments. It is also possible that 12% gels would have been more suitable to use considering the size range of the proteins detected.
4. For the Co-IP protocol, can you specify the SIGLEC-6 antibody incubation conditions with the beads?
5. In Line 167, 100 µM of pervanadate was used, while in line 197, the authors said that 5 µM of pervanadate was sufficient to see its effect on cells. Please explain what is different about the two conditions and why different concentrations were used.
6. Review protein/gene names throughout the manuscript including figure legends to make sure you follow the standard naming convention for human genes and proteins (for example, Siglec 6 gene name, Line 119 is not correct and should be SIGLEC6). Also, make sure you keep the naming consistent (i.e., hyphenated Siglec-6 vs unhyphenated Siglec 6).
7. Was a homogenizer used to pulverize the placental villi? The description in the M&M section said it was done using lysis buffer only which is insufficient to lyse the tissue.
8. Line 173, please report centrifuge speed in X g instead of rpm.
9. Line 329, remove the word endogenous considering that HTR8/SVneo does not seem to express SIGLEC-6 endogenously.
10. Are the authors sure that their cell culture incubators are at 100% humidity? Most commonly used incubators have a relative humidity of around 95%.
11. Review for the use of consistent naming, minor typos, formatting, punctuation, and grammar issues throughout the manuscript/figures.
Figure Revisions:
Figure 1:
1. Remove the red wavey underline under all the codons in Fig 1B
2. What is the difference between the blue bold and the boxed black bold letters?
Figure 2:
1. Fig 2A: no statistics seem to have been done. Please perfume a statistical analysis and indicated what type of data is plotted (i.e., mean vs median ± SEM vs SD).
2. Fig 2B: Insert a scale bar on the images.
3. Fig 2C: adjust the maginification/zoom of the membranes to match the upper and lower panel (the lower panel is more enlarged/zoomed in than the top).
4. Line 235, Fig 2A description needs correction: qPCR showing Siglec-6 expression in HTR-8/SVneo cells transfected with Siglec-6 relative to pCDH transfected HTR-8/SVneo levels (and not untransfected cells).
Figure 3:
1. Indicated how many different placentas were used to repeat this experiment (replicates using the same tissue counts as one). See major revision #1 for the issue with the blot.
Figure 4:
1. Correct the antibody label on the bottom figure panel (i.e., add phosphotyrosine).
Figure 5:
1. Was the non-specific band observed in Fig 5A membrane as well or just in Fig 5B?
Figure 6:
1. Correct the antibody label on the middle figure panel (i.e., add phosphotyrosine).
Tables:
1. I recommend changing the Table numbering in the table caption and in-text to Arabic numerals instead of Latin.

Round 2
Reviewer 1 Report
A manuscript is well revised after reviewers' suggestions.
Reviewer 2 Report
Thank you for addressing my comments. I understand, from experience, how challenging tissue work can be.